

# Monitoring newt communities in urban area using eDNA metabarcoding

Léo Charvoz[1], Laure Apothéloz-Perret-Gentil[1,2], Emanuela Reo[1], Jacques Thiébaud[3] and Jan Pawlowski[1,2,4]

[1] Department of Genetics and Evolution, University of Geneva, Geneva, Geneva, Switzerland
[2] ID-Gene ecodiagnostics, Campus Biotech Innovation Park, Geneva, Switzerland
[3] KARCH-GE (Swiss Coordination Center for the Protection of Amphibians and Reptiles)—Geneva Regional Branch, Switzerland, Geneva, Geneva, Switzerland
[4] Polish Academy of Sciences, Institute of Oceanology, Sopot, Pomerania, Poland

## ABSTRACT

Newts are amphibians commonly present in small ponds or garden pools in urban areas. They are protected in many countries and their presence is monitored through visual observation and/or trapping. However, newts are not easy to spot as they are small, elusive and often hidden at the bottom of water bodies. In recent years, environmental DNA (eDNA) has become a popular tool for detecting newts, with a focus on individual species using qPCR assays. Here, we assess the effectiveness of eDNA metabarcoding compared to conventional visual surveys of newt diversity in 45 ponds within urban areas of Geneva canton, Switzerland. We designed newt-specific mitochondrial 16S rRNA primers, which assign the majority of amplicons to newts, and were able to detect four species known to be present in the region, including the invasive subspecies *Lissotriton vulgaris meridionalis*, native to the Italian peninsula, that has been introduced in the Geneva area recently. The obtained eDNA results were congruent overall with conventional surveys, confirming the morphological observations in the majority of cases (67%). In 25% of cases, a species was only detected genetically, while in 8% of cases, the observations were not supported by eDNA metabarcoding. Our study confirms the usefulness of eDNA metabarcoding as a tool for the effective and non-invasive monitoring of newt community and suggests its broader use for the survey of newt diversity in urban area at larger scales.

## INTRODUCTION

According to the International Union for Conservation of Nature (IUCN), almost 41% of all amphibian species are threatened with extinction while 70% are drastically declining in numbers (IUCN, *Hayes et al., 2010*). The main threats faced by amphibians are habitat modification and destruction, over-exploitation of environmental resources, water and soil pollution (*Rouse, Bishop & Struger, 1999*) climate modifications and the impact of invasive alien species (*Clavero & García-Berthou, 2005*), as well as diseases (such as chytridiomycosis) (*Van Rooij et al., 2015*). Among amphibians, the newts (subfamily Pleurodelinae) seem less affected by chytridiomycosis, but their conservation status is of

Corresponding authors
Léo Charvoz,
leo.charvoz@etu.unige.ch,
leocharvoz@gmail.com
Jan Pawlowski,
Jan.Pawlowski@unige.ch

constant concern due to the destruction and pollution of their aquatic habitats. Although only two European species of newt are on the UICN Red List (*Temple & Cox, 2009*), most are protected nationally (*e.g.*, the Great Crested Newt (*Triturus cristatus*) in the UK (*Nature, 2001*; *Bormpoudakis et al., 2016*) and other European countries (*Edgar & Bird, 2006*).

Among the five species of newts present in Switzerland, only the Alpine newt (*Ichtyosaura alpestris*) is of Least Concern (*Arntzen et al., 2009*). The other four species are classified either as Vulnerable (Palmate newt, *Lissotriton helveticus*) or as Critically Endangered (Smooth newt, *Lissotriton vulgaris*; Italian crested newt, *Triturus carnifex;* and Great Crested Newt, *Triturus cristatus*). Interestingly, some species or subspecies are considered as Critically Endangered in some cantons, while invasive in others. For example, *T. carnifex* and *Lissotriton v. meridionalis* are threated in the canton of Ticino yet considered invasive in the canton of Geneva.

Because of their conservation status, newts are subject to constant monitoring. Most conventional methods for monitoring amphibians are based on visual surveys, where animals are counted in their environment. This can either be done using active techniques such as dip netting, seining or nocturnal counting by torchlight (all of which are effective for studying finite populations: *Briggs et al., 2006*; *Denton & Richter, 2012*), or passive techniques, such as traps. Since amphibians are rather timid, the use of passive techniques is known to significantly improve the efficiency of sampling (*Gunzburger, 2007*). The most effective traps will differ according to the environment in which they are set. For instance, minnow traps are appropriate for catching some amphibians during the breeding season, as they migrate to ponds and generally remain in the water until the end of the spawning period. Alternatively, pitfall traps and drift fences may be effective when placed on amphibians passageways, when animals are migrating from their foraging to their breeding habitat and vise versa (*Corn & Bury, 1990*).

Recently, the analysis of environmental DNA (eDNA) has been recognized as an efficient method for the detection of amphibian species, including newts (*Ficetola et al., 2008*; *Bálint et al., 2018*; *Goldberg, Strickler & Fremier, 2018*; *Eiler et al., 2018*). The amphibian eDNA typically consists of genetic material that is released in the environment, through mucus, secretions, excretions or other pathways (*Deiner et al., 2017*). In the case of newts, eDNA studies have largely focused on the Great Crested Newt. *Rees et al. (2014)*; *Rees et al. (2017)* demonstrated the effectiveness of this approach for detecting that species are present, both during and outside of the breeding season. Moreover, studies have also looked at the seasonal variation in eDNA detection (*Buxton et al., 2017*; *Buxton, Groombridge & Griffiths, 2018*). A further study found that the effectiveness of detection was the same using quantitative PCR (qPCR) or metabarcoding approaches (*Harper et al., 2018*). However, its application to other European newt species has not yet been tested.

Here, we use eDNA metabarcoding to survey communities of newts in the urban area of the Geneva canton, Switzerland. This area was chosen because of regular surveys targeting the invasive subspecies *L. v. meridionalis*, conducted by the Swiss Coordination Center for the Protection of Amphibians and Reptiles of Switzerland (KARCH-GE, http://www.karch-ge.ch/). The aim of this study was to assess the effectiveness of eDNA metabarcoding compared to conventional surveys and to provide complementary data

about the whole community of newts, in addition to information about the targeted invasive species.

## MATERIAL AND METHODS

### Sampling sites

Forty-five ponds were sampled in the Geneva area, Switzerland. Most of the ponds were situated in the south bank of Geneva town, in the highly urbanized area. A few sites situated in the suburbs were also examined (Fig. S1, Table S3). The sampling campaign was part of a routine monitoring survey of newt communities, organized by KARCH-GE in April 2017 at the beginning of the newt breeding period. The morphological surveys were conducted on 42 out of 45 ponds. For each pond, the number of surveys performed was often limited to one (13 sites) but in some cases the surveys were done much more frequently (10 sites), up to 100 times in the case of site 9. Newts were morphologically identified and counted in 25 out of 42 sites. At 17 sites their abundance was estimated as low, moderate or high.

### DNA barcoding

In order to develop newt-specific primers, 16S barcode sequences were obtained for five species and subspecies present in Switzerland. One specimens of *Lissotriton helveticus*, four specimens of *Lissotriton vulgaris meridionalis*, two specimens of *Ichthyosaura alpestris*, and two specimens of *Triturus cristatus/carnifex* species complex were collected in the Geneva canton, while two specimens of *Lissotriton vulgaris* were collected in the canton of Neuchâtel.

Specimens were morphologically identified, and pieces of crest or tail were preserved in ethanol and stored at −20 °C. Tissues samples were extracted using the DNeasy® Blood and Tissue kit (Qiagen, Hilden, Germany) according to the manufacturer's instructions. A fragment of 16S rRNA gene was then amplified using 16sar-L and 16sbr-H (*Palumbi et al., 1991*) with an initial denaturation at 95 °C for 5 min followed by 40 cycles of 30 s at 95 °C, 30 s at 52 °C and 45 s at 72 °C, and terminated by a final elongation step of 5 min at 72 °C. PCR products were purified using High Pure PCR Cleanup Micro Kit (Roche Kaiseraugst, Basel, Switzerland) and quantified by fluorometric quantitation using Qubit 3 fluorometer (Thermo Fisher Scientific, Ma, USA). Amplicons were then sequenced on a Sanger sequencer (ABI3130xl). Sequences were edited with CodonCode Aligner software v6.0.2 and analysed with SeaView software version 4.6 (*Gouy, Guindon & Gascuel, 2010*). Sequences were submitted to NCBI GenBank (*Benson et al., 2012*) database under the accession numbers MH818456 to MH818464 and MW418322 to MW418328 (Table S5).

### Metabarcoding primers

The DNA barcode sequences obtained in this study were aligned to 72 sequences of 8 newt species from GenBank. New primers were designed manually, taking in consideration their potential specificity to the newt species considered in this study. The forward primer 16S_1121 (5′-TTTTCCGTGCAGAAGCG-3′) shows a molecular signature on its 3′ end that appears to be shared within Salamandridae. The reverse primer 16S_1378 (5′-GCGCTGTTATCCCTAGGG-3′) is highly conserved among metazoans. The designed

primers were analyzed using Multiple Primer Analyzer online tool (ThermoFisher) to check basic parameters. Specificity of the primers was first checked *in silico* using BLAST® (*Altschul et al., 1990*). Then, the primers were tested on tissue-extracted DNA from the five newt species. To determine the resolution of newt species for the amplified fragment (alignment of 271 bp), a NJ tree (*Saitou & Nei, 1987*) was run on 79 sequences of newt species with 1,000 bootstrap replicates with algorithm implemented in Seaview software version 4.6 (*Gouy, Guindon & Gascuel, 2010*) (Fig. S1).

## Metabarcoding

### eDNA Sampling, extraction and amplification

For eDNA analysis, one litre of water was collected per site in a Nalgene sterile polyethylene terephthalate (PET) bottle (Thermo Fisher Scientific, Ma, USA). Bottles were immediately placed into a cooler filled with ice and transported to the lab, where they were stored for a few days at −20 °C until filtration.

After thawing in the dark, 750 ml of the water was filtered through Whatman Glass microfiber filters (25 mm diameter, 0.6 μm pore size) using a cleaned reusable capsule (Swinnex, Millipore). Between six and 12 filters were necessary for each site depending on the water turbidity. DNA on the filters was then extracted using DNeasy Blood and Tissue kits (Qiagen). Filters were first incubated in the lysis buffer for 48 h at 56 °C. The extraction was then performed following the manufacturer's instructions, and a final elution volume of 100 μl. DNA extracts from filter replicates per sites were pooled and stored at −20 °C until further analysis.

Extracted eDNA was amplified using the newly designed primers with an initial denaturation at 95 °C for 5 min followed by 40 cycles of 30 s at 95 °C, 30 s at 52 °C and 45 s at 72 °C, terminated by a final elongation step of 5 min at 72 °C. To ensure multiplexing of the sample into one sequencing library, tagged primers bearing eight nucleotides attached at 5′ end were included in the initial PCR reaction (*Esling, Lejzerowicz & Pawlowski, 2015*). Fifteen PCR replicates and one negative amplification control per sample were performed and replicates were pooled for further steps. Pooled PCR products were then purified using High Pure PCR Cleanup Micro Kit (Roche Kaiseraugst, Basel, Switzerland), with an elution volume of 50 μl, and quantified using Qubit 3 fluorometer (Thermo Fisher Scientific, Ma, USA). Amplified samples were pooled with approximatively 10 ng/ μl of DNA per sample for library preparation.

## High-throughput sequencing (HTS) and data processing

The library was prepared using TruSeq DNA PCR-Free kit (Illumina, San Diego, USA) following the provided protocol and quantified by qPCR using KAPA Library Quantification Kits (Roche Kaiseraugst, Basel, Switzerland). The library was finally sequenced on Illumina MiSeq System using the MiSeq Reagent Kit v2 500-cycles (Illumina, San Diego, USA). The raw sequencing data are available at the Short Read Archive public database under the accession PRJNA761253. Raw R1 and R2 fastq files for each sample were retrieved using the demultiplexer module implemented in SLIM (*Dufresne et al., 2019*). Quality filtering, removal of chimeric sequences and the amplicon sequence variant (ASV) table were generated using DADA2 R package v.1.10.1 (*Callahan, McMurdie & Holmes,*

*2017*). Species occurrence represented by less than 10 reads were not taken into account. For taxonomic assignment, IDTAXA function of the DECIPHER R package v.2.10.2 (*Wright, 2016*) was used with the local database used for DNA barcoding and a confidence threshold of 60. As the subspecies *L. v. meridionalis* could not be well distinguished from *L. vulgaris* based on the IDTAXA assignment, and *L. vulgaris* is not present in the Geneva area, all ASVs assigned to the clade *L. vulgaris/L. v. meridionalis* were assigned to *L. v. meridionalis*. The proportion of each taxonomic group was calculated after a BLAST analysis against the GenBank database with 80% of identity to the representative ASV.

## RESULTS

### Sequence data

In the DNA barcoding part of this study, 16 Sanger sequences of a fragment of the 16S rRNA gene (about 500 to 600 bp) were obtained (Table S5). Phylogenetic analysis of these sequences and other sequences of the related species available in the GenBank showed that each species formed a supported clade clearly distinct from the other species. This was confirmed by the distance tree of the short region of about 270 bp selected for metabarcoding analyses (Fig. S1). As shown by the tree, all clades are strongly supported, except the clade formed by the subspecies *L. v. meridionalis* that branches among other sequences of *L. vulgaris*.

The eDNA metabarcoding analysis was conducted on a total of 4,106,172 good-quality reads ranging from 1,836 to 184,853 reads per site. The sequences were clustered into 1028 ASVs and assigned to investigate the taxonomic composition of the amplicons dataset that was obtained with the new primers. This analysis showed that 88% of the reads belonged to the amphibians, followed by 6% to bacteria and 3% assigned to other eukaryotes (Fig. 1). Only 3% of the reads could not be assigned to any higher taxonomic group. Within amphibians, the 99.9% of the sequences were assigned to newts with a high proportion of *I. alpestris* (82%). The remaining 0.1% of the sequences were assigned to the Order Anura.

### Metabarcoding survey of newts

To investigate deeper the composition of newt populations in the Geneva urban area, the relative abundance of newt metabarcode sequences was analysed in each pond (Fig. 2). Raw data with the reference sequences as well as the assignments are given in Table S2. The distribution of newt species in the Geneva area is illustrated in Fig. 3. The map shows the locations of the 45 sampling sites and provides a zoom window on sites concentrated around the area where the invasive newt subspecies (*L. v. meridionalis* and *T.carnifex*) were first observed.

The most abundant newt species in our metabarcoding dataset was *I. alpestris*, which was present at all studied sites with a proportion varying between 1% (site ROU) and almost 100% (site 10), and reaching over 50% at 32 of the 45 sites. The second most abundant newt was *L. v. meridionalis*, detected at 53% (24) of the 45 sites, with a relative abundance reaching 95% in the case of Site 61. However, the number of reads assigned to this subspecies was small at most sites and represented by less than 10 reads in five ponds (not visible in the Fig. 2). *Lissotriton helveticus* and *T. carnifex* were both found in 11 ponds

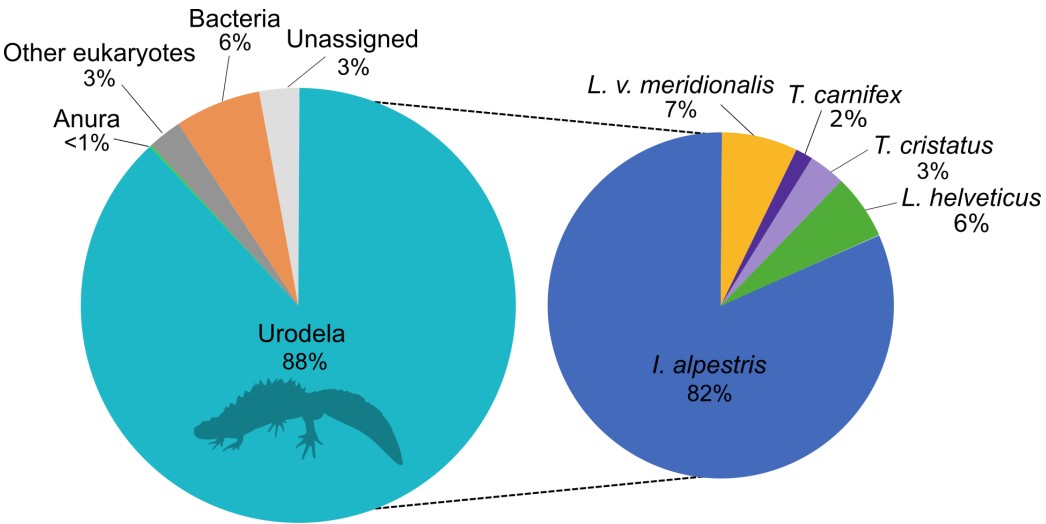

**Figure 1  Proportion of major taxonomic groups amplified by the set of new 16S primers designed in this study.** Species assignments were performed using BLAST® against the GenBank database.

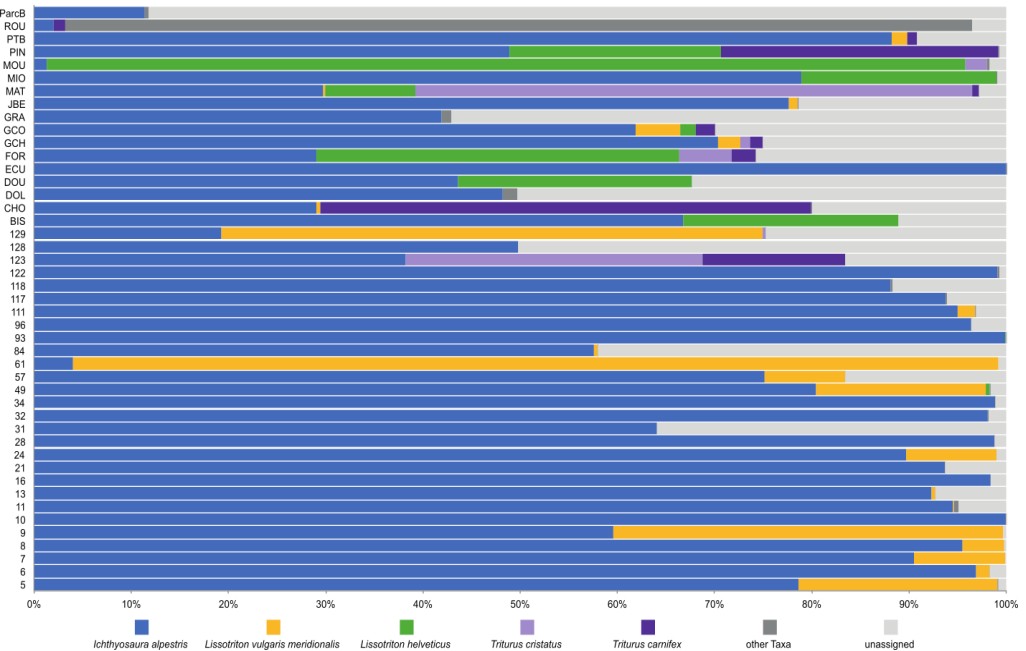

**Figure 2  Relative abundance of newts' species across 45 ponds based on metabarcoding data.**

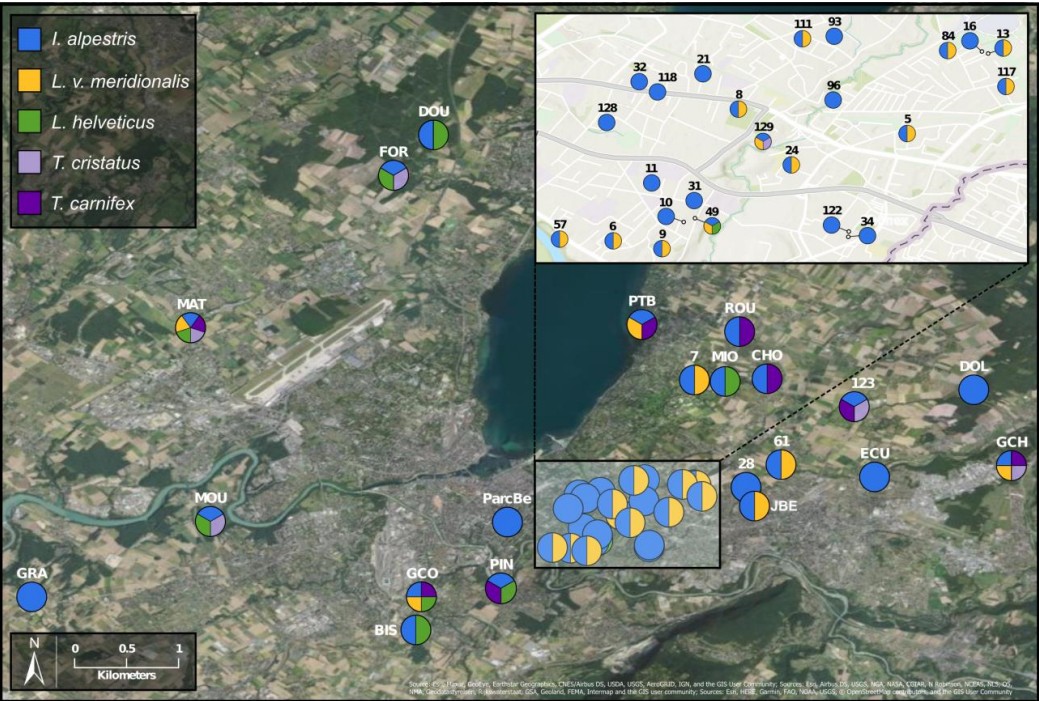

**Figure 3** **Study area and location of surveyed ponds.** Map displaying samples location in Geneva urban area. Pie charts colors correspond to the presence/absence of I. alpestris (blue), L. v. meridionalis (orange), L. helveticus (green) and T. cristatus/carnifex hybrid (purple). Map created with ArcGIS Pro on the base of the World Imagery and World Topographic Map basemaps. Source: Esri, Maxar, GeoEye, Earthstar Geographics, CNES/Airbus DS, USDA, USGS, AeroGRID, IGN, and the GIS User Community; Sources: Esri, Airbus DS, USGS, NGA, NASA, CGIAR, N Robinson, NCEAS, NLS, OS, NMA, Geodatastyrelsen, Rijkswaterstaat, GSA, Geoland, FEMA, Intermap and the GIS user community; Sources: Esri, HERE, Garmin, FAO, NOAA, USGS, ©OpenStreetMap contributors, and the GIS User Community.

with proportions reaching 95% (MOU) and 50% (CHO), respectively. Sequences of *T. cristatus* were found in 7 ponds with a maximum relative abundance of 57% (MAT), but in most of the ponds the number of reads assigned to this species was relatively small (below 10 reads in one site).

According to metabarcoding data, the co-occurrence of newt species in the same ponds was rather limited. There were only a few sites where four species were detected. At the vast majority of sites, only one or two species were detected. Remarkably, *I. alpestris* shared its habitat either with *L. v. meridionalis* or *L. helveticus*, although these two species were detected together at only one site (GCO). *Lissotriton helveticus* was often detected in the same ponds as *T. cristatus* or *T. carnifex*. The fact that the two latter species are hybridizing in the Geneva area (*Dufresnes et al., 2016*) might explain their co-occurrence at several sites.

## Comparison with morphological survey

Metabarcoding data were compared to morphological surveys for three species (*L. v. meridionalis*, *I. alpestris* and the hybrid *T. cristatus/carnifex*) that were monitored

**Figure 4** **Congruence table of the morphological and molecular presence of species across 42 ponds.** Orange color indicates presence in the morphological dataset only. Purple color indicates presence in the molecular dataset only. Green color corresponds to presence in both morphological and molecular dataset. Light green and light blue colors indicate that number of molecular reads were below 10. Single reads were ignored in this plot. Samples that were not investigated for morphological survey (samples 61, 84 and 128) were removed from the plot.

by KARCH-GE during the year 2017. The total number of observed adults and larvae at the 25 sites where the newts were counted was 2447 for *L. v. meridionalis*, 1862 for *I. alpestris* and 44 for *T. cristatus/carnifex* (Table S3). In general, the number of observed specimens increased with the number of surveys. No newts were observed at only six out of the 42 sites.

The comparison between the morphological surveys and the metabarcoding analysis regarding the presence/absence of the three newt species show that at the majority of sites, when a species was detected in morphology surveys it was also present in metabarcoding data (green in Fig. 4). There were 19 (21 considering cases with less than 10 reads) sites (blue), where a species was detected through metabarcoding alone and 6 sites (red) where a species was observed in morphological surveys but not in metabarcoding data. The most abundant species (*I. alpestris*) was detected in metabarcoding data at every site, while it was missing in morphological surveys at 6 sites. However, both sites were investigated only once during the year. In the case of *L. v. meridionalis*, the subspecies was observed morphologically and genetically in nine (11 considering cases with less than 10 reads) ponds. The nine sites that only showed a molecular signal for this species were investigated once during the year. Finally, the hybrid species *T. cristatus/carnifex* was found at 14 sites during the morphological surveys and the species was abundant at 10 sites. Metabarcoding was congruent with morphological observations in six cases (seven if the low number of reads are included). However, it failed to detect this species at five sites, of which four were indicated as having abundant population of this species. Conversely, two other sites (123, 129) showed a strong molecular signal for *T. cristatus/carnifex*, despite the fact that the species was not physically observed there.

## DISCUSSION

Our study confirms the effectiveness of eDNA metabarcoding to detect newts in aquatic environments (*Harper et al., 2018*) and to monitor freshwater pond fauna more widely (*Harper et al., 2019*). Compared to the conventional surveys, metabarcoding confirmed the presence of two species (*I. alpestris* and *L. v. meridionalis*) at all sites where they were observed. Moreover, their DNA traces were detected at an additional 15 sites where both species were not observed, suggesting that metabarcoding is more sensitive and could help to overcome the limitations of the observational approach, which is usually based on a

single observation conducted during a particular season. Interestingly, visits to some sites (*i.e.,* site 96) in April and May 2018, a year after our samples were collected confirmed the presence of the species indicated by eDNA (Karch GE, pers. comm., 2018).

There were only six cases in our study where a species was observed but not detected in metabarcoding data and all these cases concerned the crested newt complex (*T. cristatus/carnifex*). Compared to the Alpine newt (*I. alpestris*) and *L. v. meridionalis*, the crested newts were less commonly observed (14 out of 42 sites). They have been detected at only eight of these sites (including one detection based on a single read). This lack of congruence between conventional observation and metabarcoding data could be considered as a species-specific artefact, *e.g.,* due to a bias of our primers. However, this interpretation seems unlikely, as our primers fit perfectly to the two *Triturus* species, both in silico and tested on tissue samples. Interestingly, in 5 out of 6 sites where the crested newt was observed but not detected by eDNA, the specimens were not counted. Further studies of these sites might be needed to confirm whether the species was really present there.

The congruence observed in the presence/absence data was partly confirmed by abundance data. Similar agreement has been reported in several fish eDNA studies, where the number of fish eDNA reads was in rough congruence with the abundance of species (*Lacoursière-Roussel, Rosabal & Bernatchez, 2016*; *Fukaya et al., 2020*). In our study, large numbers of metabarcoding reads often corresponded to large number of specimens for the two most common newts (*I. alpestris* and *L. v. meridionalis*). The few cases where these numbers were not in agreement could be explained by the fact that the number of observations was inconsistent between sites, with some sites being inspected more often than others. Moreover, the observations were not conducted at the same time as the eDNA sampling. Another explanation could be the patchiness of newt eDNA distribution in the ponds, which might not always be encompassed the relatively limited water sampling. Although we did not expect a strong correlation between the number of reads and the abundance of specimens, our data suggest that this might be possible, at least for some species.

The main advantage of eDNA metabarcoding illustrated by our study is its capacity to survey the whole community of newts rather than a single species, the latter being the case when the qPCR-based approach is adopted (*Rees et al., 2017*; *Harper et al., 2018*). By using newt-specific primers we were able to obtain an inventory of all newt species present in the area as well as to investigate their distribution. First of all, we confirmed that the Alpine newt is the dominant species in the Geneva urban area, as it is in Switzerland as a whole. The DNA of this species was found at practically all sites, sometimes as a unique species, but more often in conjunction with other newt species. This might suggest the exceptional adaptation of this species to the life in the cities, where numerous small garden ponds offer excellent conditions for breeding (*DeTroyer et al., 2020*).

Another species that also seems to adapt easily to urban conditions is *L. v. meridionalis*. This subspecies, native to Italy, is considered as invasive in Geneva, where it was first observed in 1978 (*Jaussi, 1979*). As shown by our data, *L. v. meridionalis* often shares the breeding ponds with the Alpine newt, but it does not seem to outcompete it. In fact, we

did not find any site that was inhabited exclusively by *L. v. meridionalis*. The subspecies was also found sharing sites with crested newts (*T. cristatus/carnifex*) but only on three occasions at the periphery of its range, which might indicate that its interactions with other species are not totally neutral. According to our study, the distribution of *L. v. meridionalis* is restricted to the area delimited by southern bank of Lake Geneva, the Rhone river and the Arve river. Its presence at two sites (GCO and MAT) situated across the rivers Arve and Rhône (Fig. 3) might be explained by anthropic dispersal, which is common in urban areas. Hence, further monitoring of the expansion of this subspecies is very important, especially in view of its possible negative impact on other newts living in the area.

Globally, an encouraging result of this study is that almost all species of newts living in Switzerland could be detected in the small and highly urbanized area of Geneva canton. This finding confirms that urban areas are shelters for a wide diversity of wildlife that are adapted to its particular conditions. In this context, eDNA metabarcoding offers an efficient and reliable tool to survey this urban wildlife. Until now, metabarcoding has been mainly applied to monitoring urban fauna in terrestrial environments (*Hoffmann et al., 2018*; *Potter et al., 2019*). Its use to monitor urban aquatic biodiversity was relatively limited and focused on microbiota (*Bagley et al., 2019*; *Hervé et al., 2018*; *Hervé & Lopez, 2020*) and the detection of invasive species (*Clusa et al., 2017*), despite the fact that numerous human-made water bodies are situated in private gardens and parks. Since access to these properties is often limited, asking their owners to collect water eDNA samples might be easier to organize than conventional observations. The non-invasive collection of large amount of data for routine monitoring has obvious practical advantages, and would also contribute to raising interest in urban biodiversity and motivating residents to protect it.

## CONCLUSIONS

Our study confirms the usefulness of eDNA metabarcoding for monitoring aquatic biodiversity in urban areas. Taking the newts as an example, we show that the urban ponds are inhabited by a rich community of species. We found a good congruence between eDNA data and conventional observations. Yet, the detection of some newts' species seems easier than the others. Further studies are needed to evaluate the impact of abundant species on the detection of rare species in eDNA datasets. This and other issues related to eDNA data interpretation can only be solved by more regular eDNA surveys, which will hopefully follow this precursor study.

## ACKNOWLEDGEMENTS

The authors are thankful to KARCH-GE Team: Emeline Chapron, Lucien Guignet, as well as to the owners of examined ponds and the authorities. We also thank to the three anonymous reviewers for very thoughtful comments and to Prof. Andrew Gooday (NOC, Southampton) for correcting the English.

### Funding

This work was supported by the University of Geneva. The funders had no role in study design, data collection and analysis, decision to publish, or preparation of the manuscript.

### Grant Disclosures

The following grant information was disclosed by the authors:
University of Geneva.

### Competing Interests

Laure Apothéloz-Perret-Gentil and Jan Pawlowski are employed by ID-Gene ecodiagnostics, Switzerland.

### Author Contributions

- Léo Charvoz conceived and designed the experiments, performed the experiments, analyzed the data, prepared figures and/or tables, authored or reviewed drafts of the paper, and approved the final draft.
- Laure Apothéloz-Perret-Gentil conceived and designed the experiments, analyzed the data, prepared figures and/or tables, authored or reviewed drafts of the paper, and approved the final draft.
- Emanuela Reo conceived and designed the experiments, performed the experiments, authored or reviewed drafts of the paper, and approved the final draft.
- Jacques Thiébaud conceived and designed the experiments, prepared figures and/or tables, and approved the final draft.
- Jan Pawlowski analyzed the data, authored or reviewed drafts of the paper, and approved the final draft.

### DNA Deposition

The following information was supplied regarding the deposition of DNA sequences:
The DNA sequences are available at GenBank: MH818456 to MH818464 and MW418322 to MW418328.

### Data Availability

The raw data is available in the Supplementary Files.

### Supplemental Information

Supplemental information for this article can be found online at http://dx.doi.org/10.7717/peerj.12357#supplemental-information.

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
