# Peer review of "Monitoring newt communities in urban area using eDNA metabarcoding"

_PeerJ, doi:10.7717/peerj.12357_

## Round 0.1 · original submission · Minor Revisions

After having read the ms and the comments from the reviewers, I find that this ms will be acceptable if the authors follow the recommended changes:

1. The aims of this study are somewhat obscure, and do not match the data that was available prior to the study. Notably, the introduction does not mention that the sites were all known to be positive for the target taxon - meaning that this was an exercise of confirmation rather than initial determination (as per rev #2). Also, at L159-62, please make it clear that all ponds sampled are known to have populations on newts.

2. I question the assertion (L118-9 & L253-4) that all Triturus cristatus and T. carnifex should be treated as hybrids (contrast with your results at L242-4). Surely, your eDNA technique (mtNDA) does not allow you to distinguish between pure parental types and hybrids. Thus, to assume hybrids for all just because they can hybridize is faulty logic. I'd suggest referring to this as a complex as you cannot distinguish what you are sequencing.

3. Provide full data on your negative controls (as per rev #3).

Please note that review time for this work was significantly increased by the need for reviewers to spend time on the poor English of the submission. Although it is recognised that the authors are not from a English speaking region, several authors are known to produce other works fluently, suggesting that it would have been possible to improve the quality prior to submission. I trust that the authors will not only amend the English corrections provided by the reviewers, but make a thorough and sincere overhaul of their work for readers.

Reviewer 1 ·

Basic reporting

The paper is interesting and globally well-written. Especially, this survey method is very useful since the habitat fragmentation occurs in a context of global rapid urbanization.

Experimental design

This is a relevant manuscript that perfectly fits the aims and scope of "PeerJ".

Validity of the findings

no comment

Additional comments

1.The Materials and Methods section should include a description of the study area.
2.The figures in the article are not clear. I recommend using vector graphics.

Reviewer 2 ·

Basic reporting

Unfortunately, the manuscript requires significant edits to make it suitable for publication. I can appreciate that English may not be the authors first language, and where possible, I have tried to help resolve grammatical issues. My main concerns involve the structure of the manuscript, the lack of clarity in several areas, the failure to include key figures (such as a map of the study region), and the failure to incorporate all citations in the reference list. I have made several comments in the document with respect to this. With respect to the map, I realise one is provided in Fig 3, but it reflects the results of the eDNA work. A lot of my uncertainty in the design may have been alleviated by a clearer figure to better orient me to the general region.

In particular, I would like to emphasize that it was unclear what was done for this study and what had been done in previously published works. Was the barcoding work specific to this study? No sample collection was made explicit; only GenBank numbers. The main sampling seems to have been for the primer design (which was still vague) and the eDNA work.

Within your introduction, the specific aims and hypothesis of your study were not stated. Nor was any substantial mention given to your study region and why it was chosen or appropriate. It would seem that one species of subspecies could be invasive in certain ponds; however, the importance of detecting invasives and that this was possibly a key aim of this study was never mentioned. Because of this the introduction seemed broken up and ill-structured.

There were also instances where content presented in one section would be more suited in another. For example, parts of the results should be in the methods, parts of the methods in the introduction, etc.

I also encourage you to better integrate your figures into your statements. In many instances it reads as though the figure caption is simply pasted into the results, for example. Try to make your narrative flow, and cite your tables and figures where appropriate to help enhance your point.

Experimental design

Overall, I understand the general eDNA approach used by the authors and I perceived their aims to assess the effectiveness of the eDNA approach compared to demographic surveys, and to assess the distribution of 5 newt species in the Geneva canton, including assessing the extend of some invasive species. Unfortunately, these were not explicitly stated. As such, I may have misinterpreted things.

The first components of their methods was not provided in sufficient detail. I was confused as to whether the reference library (DNA barcoding) was performed by them or if this was previously published. GenBank numbers were provided for 4 sequences, but the authors report generating 16 sequences. Also, the development of the primers was not provided in adequate detail. Perhaps a more comprehensive account of this could be provided in Supplementary material if it would not fit within the world limit.

Validity of the findings

The validity of the findings are weakened by the fact that in some instances, the methods applied were not consistent. Also, key aspects of the discussion may be better suited as the background in the introduction.

With that said, I do believe there may be valid findings here. The paper simply needs to define clearer objectives and remain focused to these.

Annotated reviews are not available for download in order to protect the identity of reviewers who chose to remain anonymous.

Reviewer 3 ·

Basic reporting

no comment

Experimental design

no comment

Validity of the findings

no comment

Additional comments

This was an interesting manuscript to review. The manuscript is well-organized and describes an excellent application for eDNA metabarcoding to detect and quantify newt assemblages in urban aquatic habitats. I have some minor questions and comments, but these should be considered as food for thought and not essential revisions. My only substantive comment is that I would encourage the authors to have a colleague whose first language is English to go through the manuscript to improve the grammar and wording, as this will improve the manuscript’s reader uptake and impact.

The organization and content of the manuscript are sound, and describe a novel and well-executed study. My only substantive suggestions are to include more information on what negative controls were included in the eDNA sampling and processing steps (e.g. field negative controls, likewise filtering, extraction, and amplification negative controls) and to improve the wording to make the manuscript easier to read (see minor comments below).

If the authors wanted to take their results further, it would be interesting to quantify alpha and beta diversity within and among the sampled waterbodies, particularly since these are primarily urban and near-urban habitats. This is just a suggestion, however, as it was not a stated objective of the manuscript.

Minor comments:
l. 32 – change “Since few years” to “In recent years”
l. 33 – change “became” to “has become”
l. 34 – I’d suggest changing “with focus on the detection of great crested newt using qPCR assays” to “a focus on detecting individual species of interest rather than newt assemblages or species richness”
l. 41 – change “the species” to “a species”
l. 42-43 – please reword this sentence to improve clarity
l. 45 and throughout the manuscript– change “Geneva area” to “the Geneva area”
l. 48 – change “newts diversity in urban area at larger scale” to “newt diversity in urban areas at larger scales”
l. 57 – remove “factors” (redundant wording)
l. 63 – add “IUCN” before “red list” for clarity
l. 87 – change “On the opposite,” to “Alternatively,”
l. 91-92 – reword the first part of this sentence to read “Recently, analysis of environmental DNA (eDNA) has been formally recognized as an efficient method…”
l. 97-98 – reword the first part of this sentence to read “In the case of newts, eDNA studies have largely focused…”
l. 103 – reword the end of this sentence to read “…newt species has not been tested”
l. 106 – change “use” to “used”
l. 108 – remove “it was” and “the”
l. 117 – change “later” to “latter”
l. 119 – change “therefore are considered as hybrids” to “are therefore considered as potential hybrids”
l. 120 – remove “and have been”
l. 125 and elsewhere – the 16S rRNA mitochondrial region is not a gene; better to refer to it as a region
l. 145-146 – change “the primers pair” to “the primer pairs”; also change “the five different newts DNA extracted from tissues samples” to “tissue-extracted DNA from the five newt species”
l. 146-150 – please include references to support these statements
l. 157 – change “campaign was performed” to “was carried out”, “was conducted”, or something similar; also change “the beginning of newts” to “the beginning of the newt”
l. 159 – add a space between Table and S1
l. 161 – change “25 of ponds” to “25 of the sampled ponds”
l. 162 – add “a” before “Nalgene”
l. 165 – remove “step”
IMPORTANT: please include information on what negative controls were included in the eDNA sampling and processing steps (e.g. field negative controls, likewise filtering, extraction, and amplification negative controls)
l. 166 – add “the” before “dark”; also replace “on” with “through”
l. 171 – change “extract” to “extracted”
l. 174 – change “Water eDNA samples were then amplified” to “Extracted eDNA was amplified”
l. 177-178 – were index tags included in the initial PCR reaction, or added in a secondary PCR reaction? Please clarify this
l. 179 – please provide more information on the negative control; presumably this was an amplification negative control?
l. 192 – data not available to check
l. 194 – change “chimera” to “chimeric”
l. 221 – add “the” before “GenBank database”
l. 228 – if using formal names, this should be “the Order Anura”
l. 234 – change “newts metabarcodes” to “newt metabarcode sequences”
l. 240 – remove “the”
l. 241 – remove “of”
l. 247 – change “newts species” to “newt species”
l. 249 – remove “the” after “Remarkably”
l. 251 – remove “the” before “site GCO”
l. 252 – change “later” to “latter”
l. 253 – change “capable to hybridize in Geneva area” to “hybridizing in the Geneva area”
l. 253 – change “at Fig. 3” to “in Fig. 3”
l. 268 – change “In the case of 25 sites” to “At 25 sites”
l. 270 – add “the” before “25 sites”
l. 274 – add “sites” after “42”
l. 275 – remove “the” at the end of the line
l. 280 – change “the species” to “a species”
l. 284 – include this information in the Methods, since it has a large effect on what’s filtered out before being considered data
l. 285 – add brackets around “I. alpestris”
l. 288 – remove the comma after “9 sites”
l. 292 – clarify whether the result shown in light green should or should not be included (ie decision criteria – this should be stated in the methods)
l. 296 – add “physically” before “observed there”
l. 304 – similar comment on inclusion criteria for observations shown in light blue
l. 309 – who is the “pers. comm.” From (name and affiliation)?
l. 331 – change “newts” to “newt”
l. 348 – add a comma after “Italy”
l. 361 – change “the amazing result” to “an encouraging result”. Saying the result is “amazing” would contradict the authors’ arguments for using eDNA in the Introduction, as well as the growing literature that documents the effectiveness of eDNA detection
Figure 2 – change “newts’ ” in the figure caption to “newt”
Figure 3 – please state what the enclosed areas of the figure show

---

## Round 0.2 · accepted · Accept

Thanks for your revision of this manuscript. I am now happy with it to be accepted for publication in PeerJ.

Please note some minor editorial issues on the attached - especially the grammatical error in the title!